# Conservation Environments’ Effect on the Compressive Strength Behaviour of Wood–Concrete Composites

**DOI:** 10.3390/ma15103572

**Published:** 2022-05-17

**Authors:** Walid Khelifi, Selma Bencedira, Marc Azab, Malik Sarmad Riaz, Mirvat Abdallah, Zaher Abdel Baki, Andrey E. Krauklis, Hani Amir Aouissi

**Affiliations:** 1Laboratory of Civil Engineering, Department of Civil Engineering, Faculty of Technology, UBMA, Annaba 23000, Algeria; khelifi.walid23@gmail.com; 2Laboratory of LGE, Department of Process Engineering, Faculty of Technology, UBMA, B. P12, Annaba 23000, Algeria; 3College of Engineering and Technology, American University of the Middle East, Kuwait; marc.azab@aum.edu.kw (M.A.); mirvat.abdallah@aum.edu.kw (M.A.); zaher.abdelbaki@aum.edu.kw (Z.A.B.); 4Civil Engineering Department—National University of Technology (NUTECH), Islamabad, Pakistan; sarmadriaz@nutech.edu.pk; 5Institute for Mechanics of Materials, University of Latvia, Jelgavas Street 3, LV-1004 Riga, Latvia; andrejs.krauklis@lu.lv; 6Scientific and Technical Research Center on Arid Regions (CRSTRA), Biskra 07000, Algeria; aouissi.amir@gmail.com

**Keywords:** renewable materials, wood cuttings, vegetal fibre concrete, compressive strength, mechanical behaviours, experimentation

## Abstract

This paper addresses the issues in making wood–concrete composites more resilient to environmental conditions and to improve their compressive strength. Tests were carried out on cubic specimens of 10 × 10 × 10 cm^3^ composed of ordinary concrete with a 2% redwood- and hardwood-chip dosage. Superficial treatments of cement and lime were applied to the wood chips. All specimens were kept for 28 days in the open air and for 12 months in: the open air, drinking water, seawater, and an oven. Consequently, the compressive strength of ordinary concrete is approximately 37.1 MPa. After 365 days of exposure to the open air, drinking water, seawater, and the oven, a resistance loss of 35.84, 36.06, 42.85, and 52.30% were observed, respectively. In all environments investigated, the untreated wood composite concrete’s resistance decreased significantly, while the cement/lime treatment of the wood enhanced them. However, only 15.5 MPa and 14.6 MPa were attained after the first 28 days in the cases of the redwood and the hardwood treated with lime. These findings indicate that the resistance of wood–concrete composites depends on the type of wood used. Treating wood chips with cement is a potential method for making these materials resistant in conservation situations determined by the cement’s chemical composition. The current study has implications for researchers and practitioners for further understanding the impact of these eco-friendly concretes in the construction industry.

## 1. Introduction

The construction sector directly contributes to the expansion of economic activities while, at the same time, it is a major consumer of natural and physical resources. Unfortunately, this expansion has adverse environmental, social, and economic impacts. The environmental impact of construction activities has increased in recent decades due to the rapid increases in the population and greater industrial activity [1,2]. The building industry is responsible for around a quarter of all CO_2_ emissions [3], not to mention the loss of non-renewable resources [4]. As a result, the building industry must innovate to ensure customer satisfaction, while also being environmentally sustainable, hence the interest in creating an eco-construction technique [5,6,7]. This unique design technique strives to produce pleasant structures at a reduced environmental cost by using more efficient materials [8]. In this context, agro-materials, partially produced from biomass [9], are rapidly being developed and commercialized in the construction materials market [10,11]. Because of their ecological nature, these agro-materials make it possible to enhance the environmental balance of construction, particularly of buildings [12,13,14,15]. Among the agro-materials, vegetable concretes have been designed to be used in the construction industry [16,17] to take advantage of their thermal, acoustic, and hygroscopic qualities [18,19]. These concretes generally encourage the recovery of by-products from different economic sectors. Academics and practitioners have shown interest in investigating and comprehending these materials in the presence of various stressors [20]. Plant or cellulose fibres (CFs) are the most-often utilized vegetable concretes [21,22] as they are characterized by low cost, light weight, good adhesion, a simple production method, and biodegradability, which is attracting the attention of an increasing number of researchers.

Cellulose fibre (CF) is one of the world’s most plentiful natural resources, and it may be found in a variety of agricultural leftovers, including rice straw, rice husk, maize straw, bagasse, wood shavings, wood chips, bamboo chips, and so on [23,24]. These agricultural wastes are mostly made up of cellulose, hemicellulose, lignin, pectin, wax, and other water-soluble components [25]. CF is essentially identical, with slight density variations between 1.1 and 1.6 g cm^−3^. For example, softwood has a density of roughly 1.5 g cm^−3^, a reasonably high tensile strength, and a comparatively low tensile strength compared to fruit coconut-husk fibre [26]. Most of this field’s research has focused on cementitious wood-composite panels based on pervious concrete. Cement-bonded–wood-fibre composites are not new in the building industry [27,28]. These materials are less harmful to the environment and less expensive to manufacture. They have a thermal conductivity similar to expanded polystyrene and glass wool [29,30]. In addition, fibre-based composite materials have sparked fresh attention because of their potential to adjust interior climates [31]. Numerous investigations have demonstrated that the hygroscopic behaviour of materials of vegetable origin allows for the regulation of ambient humidity [32]. In addition, they can be easily employed for both new construction and renovations [33]. They also offer additional benefits, such as good phonic insulation due to their ability to absorb sound waves [34,35,36]. Another significant advantage of using wood filler in cementitious materials is its cost; it is readily accessible, ecologically acceptable, and simple to process [37,38,39]. However, the primary challenges faced throughout numerous studies of wood–concrete products have been the compressibility of the wood particles inside the hardened concrete due to their low level of stiffness [40] and the incompatibility induced by soluble chemicals during the fresh cement phase [41]. In fact, they can raise the pH of the cement mixture over 12.5 due to the creation of Ca(OH)_2_, which aids in dissolving low-molecular-weight compounds inside the wood particles. As a result, the wood particles must be treated before being employed in the concrete composite [42]. Table 1 summarizes the many ways generally used for the treatment of wood. The coating technique is a typical treatment method for minimizing the influence of soluble chemicals while also boosting volume stability and surface roughness [43].

Thus, an experimental approach, developed by Khelifi et al. [49], was performed to assess the thermo-mechanical properties of treated/untreated wood-chip-concrete. The utilised redwood and hardwood chips were obtained from carpentry waste, namely trash from routing and planning work. The utilised redwood and hardwood chips were treated with cement and lime. The experimental research findings for the concrete properties demonstrate the higher performance of the examined concrete compared to composite concrete made with untreated wood chips. The treatment enhances the adhesion of the wood matrix. Ordinary concrete can benefit from the addition of wood chips to improve its heat conductivity. This improvement of construction material made from renewable resources has the potential to provide high thermal insulation. However, until specific issues impeding their utilisation are addressed, the use of wood–concrete materials in the building industry will remain limited. In addition, all of the data were obtained just after the material’s cement hardened (after 28 days in the open air). If they are to be employed as structural elements, their mechanical properties must be improved. They must also withstand the conditions imposed by the environments in which they will be utilized for as long as possible. Due to the complexities of the effects involved in this investigation, only compressive strength will be evaluated. Finally, this research will fill the gap in the literature regarding the effect of adding wood chips to concrete paste by addressing the following questions:How does ordinary concrete compressive strength behave in storage environments?What effects do treated/untreated wood chips have on this behaviour?Does the type of wood influence this behaviour?How does the chemical composition of the cement used in wood-chip treatments contribute to the composite’s resistance to various storage conditions?

## 2. Materials and Methods

### 2.1. Research Methodology

Figure 1 depicts the itemized, strategic system for the development of the compressive strength investigation of cement materials made from redwood and hardwoods. After being preserved in multiple environments, specimens underwent compressive strength studies. Those specimens were made of ordinary concrete and treated/untreated wood–concrete composites. There were two types of wood used: redwood and hardwood. The treatment of wood chips can take several forms; in this investigation, the wood chips were coated with cement and lime. The duration and the conservation environments were as follows: 28 days in the open air and then 365 days in: (i) open air, (ii) seawater, (iii) drinking water, and (iv) an oven.

The steps mentioned above were followed to develop the ordinary concrete and the treated/untreated-wood concrete using established concept development and strength analysis.

### 2.2. Materials

The shavings were by-products of woodworking processes. Redwood (W_I_) was produced from the wood of a Scots pine tree. According to NF EN 1611 European classification standards, W_I_ is classified as VI Scandinavian. Hardwood (W_II_) was steamed beech sawn from Romania. European standard NF EN 975-1 classifies W_II_ as a pedunculated oak species. Both of them have rapid growth and are cost-effective; these species of wood are widely employed in the construction sector. Both are distinguished by rapid development and low-cost aggregates. W_II_ has higher mechanical properties than W_I_.

The density at RH 15% is typically 0.1 for redwood and 0.3 to 0.4 for hardwood. Both, measuring 0.5–12.5 mm, had an irregular form [49]. It should be ensured that the chips are evenly dispersed in the cement matrix throughout the mixing process. 

The multiple grain sizes that make up a sample may be determined and seen via granulometric analysis. The granulometric analysis is performed using a LATEST SPA sieve machine set to moderate vibration. The study entailed sorting and categorising the grains based on their diameter using sieves piled one on top of the other, with the diameters of the holes decreasing from top to bottom. In general, the sample analysed in this study was placed on the top sieve, and the categorisation of the grains was acquired by the vibration of the sieve column. The column was stirred mechanically. The sieves were then agitated one by one, starting with the one with the widest opening, and using a bottom and a cover. When the residue did not change by more than 1% after one minute of sifting, the sieving was complete. The refuse was passed through each sieve, and the results were added together. Figure 2 shows the chip granulometric curves of W_I_ and W_II_. It clearly shows that the wood chips had an irregular form, a higher granulometric limit of approximately 12.5 mm, and a lower granulometric limit of approximately 0.5 mm. Because of the chips’ lack of stiffness and geometry, none of these characteristics was of any relative value.

The method appeared to be adequate for supplementing scanning electron microscopy (SEM) of surface features. Therefore, SEM analyses of wood-chip samples were performed using a JEOLJSM100F spectrophotometer. Figure 3 and Figure 4 show SEM microstructural images of W_I_ and W_II_ for various view fields ranging from 10 µm to 500 µm. From Figure 3, we can see that W_I_ chips have a fibrous texture with ordered distribution, while Figure 4 shows the ordered distribution of W_II_’s fibrous texture. Despite this fact, the W_I_ and W_II_ are rather compact materials with fibres that run in the same direction.

The geometrical dimensions of the wood chips may also be seen in the SEM pictures. According to SEM examination, the wood chips had a low recovery property.

Portland cement (type II) class 45 (CPJ-CEM II/A) was utilized with the following physical characteristics: a specific surface area of 289 m^2^ kg^−1^, a specific density of 3078 kg m^−3^, and a loss on ignition of 1.16%. Table 2 and Table 3 show the mineralogical and chemical compositions of the cement, respectively. The Bogue method was used to determine both of these at the Hadjar Essoud (Skikda, Algeria) cement plant laboratory. Table 2 shows the chemical compositions as determined by the Hadjar Essoud [50] (Skikda, Algeria) cement plant laboratory (SCHS), from which the used Portland cement was obtained. On the other hand, Table 3 shows the mineralogical composition of the cement. It is derived from the data in Table 2 using the Bogue calculation [51]. In fact, the Bogue method was used to approximate the proportions of the four primary minerals in the Portland cement clinker [52]. Despite the fact that the results are only approximate, the computation is incredibly valuable and commonly used in the cement industry. The calculation assumes that the primary clinker minerals are pure minerals with the following compositions: (i) alite or tricalcium silicate (C_3_S); (ii) belite or dicalcium silicate (C_2_S); (iii) tricalcium aluminate (C_3_A); and (iv) tetracalcium aluminoferrite (C_4_AF) [53]. These clinkers may be computed using Equations (1)–(4) [54] where the oxides reflect the weight percentages (PW) [53]:C_3_S = 4.0710 CaO − 7.6024 SiO_2_ − 1.4297 Fe_2_O_3_ − 6.7187 Al_2_O_3_(1)
C_2_S = 8.6024 SiO_2_ + 1.0785 Fe_2_O_3_ + 5.0683 Al_2_O_3_ − 3.0710 CaO(2)
C_3_A = 2.6504 Al_2_O_3_ + 1.6920 Fe_2_O_3_(3)
C_4_AF = 3.0432 Fe_2_O_3_(4)

The concrete mixture contained 3 types of aggregates: rolling sand (2), fine aggregate (1), and coarse aggregate (3). Their mechanical and chemical characteristics are summarized in Table 4 and Table 5 [49]. The aggregate characterization techniques are outlined by [55,56,57,58].

### 2.3. Preparation and Conditioning of Test Specimens

#### 2.3.1. Wood Chip Treatment

Regarding the treatment of the shavings, many treatments have been utilized in earlier studies, and it has been demonstrated that treatment by coating lime with cement yields satisfactory results, particularly in terms of mechanical strength and shrinkage [59,60]. Indeed, before adding these two types of wood chips to the composition, they were immersed in the combinations (cement + water) and then (lime + water). Lime milk was made from natural hydraulic lime (NHL5). The treated chips were left to dry in their natural state for more than 3 days before use. Indeed, for cement treatment of the W_I_ and W_II_ surfaces, they are immersed individually in cement milk or in water. The cement and lime used in this operation were Portland cement and natural hydraulic lime (NHL5). Similarly, the two types of wood were immersed in cement milk for the lime treatment. Cement and lime milks are cement–water and lime–water mixtures, respectively. Both of them were prepared, respectively, with water–cement (W/C) and water–lime (W/L) ratios equal to 1. After complete saturation, the treated chips were allowed to dry in the open air for 3 days before being used. Figure 5 shows the W_I_ and W_II_ before and after the cement and lime treatments. 

Seven different types of specimens will be discussed in this article:OC: Ordinary concrete,UW_I_: Untreated redwood–cement concrete,UW_II_: Untreated hardwood–cement concrete,TW_Ic_: Treated redwood–cement concrete (with cement),TW_IIc_: Treated hardwood–cement concrete (with cement),TW_IL_: Treated redwood–cement concrete (with lime),TW_IIL_: Treated hardwood–cement concrete (with lime).

#### 2.3.2. Formulation of Concrete

Previous research [49] has provided the ideal compositions of the sand concrete examined without including wood shavings. Indeed, concrete is a glue-based material that holds the fillers together (fine aggregates, coarse aggregates, rolling sand, and wood chips). The paste and fillers combine to form the heterogeneous substance known as “concrete.” The Dreux–Gorisse technique [33] is used to investigate the concrete composition. The compositions shown in Table 6 were used to prepare the mixtures. Figure 6 shows the results of a particle size study performed on the aggregates. The aim of this formulation was to generate a common concrete with a characteristic compression resistance of 30 MPa at 28 days. The maximum diameter (D_max_) was set at 1, the average vibration speed was 1, and the proportion of aggregates was as follows: 35% rolling sand, 10% fine particles, and 55% coarse aggregates (3). The compositions shown in Table 6 were used to prepare the mixtures.

#### 2.3.3. Concrete Mixing

These procedures are sensitive because they are required to assure the status of the coating and the holding of the chips. Therefore, a well-defined procedure and low-speed mixing (about 50 turns mn^−1^) were adopted. The dry sand, cement, gravel, and wood chips were combined for 1 min at a slow speed. When the mixture was entirely homogenous, the mixing water was progressively added at a slow pace for 4 min. Using a vibrating table at a rate of 50 Hz and a vibrating time of 1 min resulted in the separation of distinct components. Finally, the moulds were manually filled.

#### 2.3.4. Preparation of Specimens

The slump test for concrete was assessed using an Abrams cone after it had been mixed (AFNOR P 18-451). The Abrams cone slump was used to calculate handling. The new concrete was introduced into a cubic mould (10 × 10 × 10 cm^3^) on a vibrating table for 1 min. The specimens were stored in an appropriate location for 24 h and then immersed in water according to the NFP18404 standard.

### 2.4. Specimen Storage

All of the types of specimens were kept in an appropriate area for 1 day before being submerged in water for 28 days, according to the NFP18404 standard. Following that, 4 series of test specimens were stored for 12 months in a variety of conservation environments:Seawater (conditions: T = 25 °C, RH 100%, pH 7–9),Drinking-water (conditions: T = 25 °C, RH 100%, pH 6–7),Open-air (conditions: T= 25 °C, RH 55%, pH 12–14),Oven (conditions: T = 75 °C, RH < 50%, pH 12–14).

### 2.5. Experimental Methodology of Compressive Strength Test

Each reported result for the compressive strength tests was the average of 105 samples. The seven categories of samples were tested for each series. Each kind was tested three times to ensure that the results would be reproducible. As a consequence, the total number of specimens used for each series was 21. The dry compressive strength was tested using a Baratest AG hydraulic press built in Switzerland; the compressive strength loss (RL) and gain (RG) are computed using Formula (5):(5)$$\mathrm{RL}\left(\%\right)=\left[\frac{{R\;-\;R}_{\mathrm{SC}}}{R}\right]\times 100\;$$

R_SC_ is the control specimen’s compressive strength (MPa), and R is the compressive strength of a specimen of the same type in each environment (MPa). 

### 2.6. Density Determination

Density was calculated by multiplying the mass of each specimen and its envelope volume, as shown in Equation (6), where M_0_ is the weight (g), V_0_ is the volume (cm^3^), and is the density (g cm^−3^). The average of five measurements was used to calculate the results.
(6)$$\rho =\frac{m_{0}}{V_{0}}\;$$

### 2.7. Water Absorption Capacity Measurement

The water absorption capacity was calculated using ASTM D 1037-12. A total of 6 specimens were submerged in water at a temperature of 20 °C for each sample. After 2 h and 24 h, the mass measurements were obtained. Each specimen’s water absorption was determined as a percentage of mass gain compared to the starting mass, as shown in Equation (7) where m_sat_ (g) is specimen mass in a saturated state and m_dry_ (g) is the specimen mass in a dry state:(7)$$\omega =(\frac{m_{\mathrm{sat}\;}{-\;m}_{\mathrm{dry}}}{m_{\mathrm{dry}}})\;\;$$

### 2.8. XRD Characterization

A Rigaku Ultima IV multipurpose X-ray diffraction apparatus was used to analyse the crystalline structure of the treated and untreated wood chips. The diffractometer was equipped with an X-ray generator (maximum rated output: 3 kW, target: Cu) and a goniometer (maximum rated output: 3 kW, target: Cu). The high score application, in accordance with the International Centre for Diffraction Data (ICDD) PDF-4, was used to identify the phases and retrieve all of the crystalline structure characteristics.

## 3. Results and Discussions

### 3.1. Ordinary Concrete

From the findings in Figure 7, RLs of the OCs were evident in all media, and they depended on the conservation context. The compressive strength of the control OC was about 37.1 MPa. The literature shows a mean compression strength of 37.3 MPa in moulded cement composites made using Portland cement and no wood [61]. The creation of glue-water-cement explains this result during the first 28 days of concrete setting. Le Chatelier (1887) [54] described a cement hydration process based on dissolution/precipitation. When water is supplied, the anhydrous reactants will gradually dissolve. These reactants are more soluble than the products after hydration. The solution becomes supersaturated, resulting in the precipitation of hydrates. As the concentration of reactants (Table 3) falls, more ions enter the solution, allowing other hydrates to precipitate. The primary phases of Portland cement are tricalcium silicates (C_3_S) and dicalcium silicates (C_2_S) (Table 2). Their hydration products are hydrated calcium silicates (C-S-H) and Portlandite (Ca(OH)_2_) [62,63].

Figure 7 depicts RLs of 35.84%, 36.06%, 42.85%, and 52.30%, after 12 months of exposure to outdoor air, drinking water, seawater, and an oven, respectively. Hardened concrete improves its compressive strength over time under normal conditions. However, it should be noted that environmental variables, particularly RH, temperature, and inorganic salts, might influence the evolution of compressive strength.

The evaluation of the impact of RH on the compressive strength of fresh and hardened cement is based on a concerted effort and is strongly connected to the phenomena of carbonation [64]. Carbonation, according to Mahmood et al. [65], is described as a regularly diffused phenomenon in ancient reinforced concrete structures, which are typically constructed without special design requirements preventing degradation. In the carbonation process, atmospheric carbon dioxide (CO_2_) enters the concrete cover through the capillary pores and reacts with calcium hydroxide (Ca(OH)_2_), which is a hydration product of calcium silicate compounds present in the clinker and which is responsible for pH value reduction, resulting in the formation of calcium carbonate, as depicted by Equation (8):Ca(OH)_2_ + CO_2_ → CaCO_3_ + H_2_O(8)

Furthermore, CO_2_ may react with the calcium silicate hydrate (C-S-H) network to produce additional CaCO_3_ [66]. The proportion of Portlandite (Ca(OH)_2_) in the concrete mass determines the pH of the recycled aggregate. Because of the dissolution of Ca(OH)_2_ and the generation of OH- in water, the pH of the recycled aggregate increases to 12 and normally ranges from 11 to 13. According to Equation (9), the presence of water in the carbonation process is required for the mobility of hydrated cement products in the pore solution and allows CO_2_ to dissolve and create carbonic acid (H_2_CO_3_). Using the following process, this weak acid attacks Portlandite (Ca(OH)_2_) and generates CaCO_3_:Ca(OH)_2_ + H_2_CO_3_ → CaCO_3_ + 2H_2_O(9)

Elsalamawy et al. [67] established a direct proportionality between carbonation depth and concrete compressive strength. This is supported by Peng Liu et al. [25], who reinforced this conclusion, noting that cement type has a significant impact on concrete carbonation resistance. Metalssi et al. [67] report that, for genuine constructions exposed to natural carbonation, the concrete is saturated after demoulding and cannot be carbonated spontaneously owing to the low CO_2_ content. Carbonation, on the other hand, might occur after a partial drying of the material, after several weeks, or even after several years (in the case of Portland cement, CEM I according to European standards) [68].

Elsalamawy et al. [67] demonstrate that RH is one of the most important elements influencing carbonation depth, with the carbonation depth increasing with increasing RH to reach a peak value at 65 percent RH, regardless of cement type. This discovery validates the importance of the electrified RL for open-air OC preservation (conditions: RH = 65%, T = 25 °C, pH 12–14).

In addition, the same authors [67] determined relative carbonation depth by dividing the value of carbonation depth by the maximum value of carbonation depth. They established the polynomial relationship between relative carbonation depth and relative humidity (40–80%). The graph demonstrates that at RH 50%, there is insufficient moisture for carbonation to occur, and at RH > 70%, there is insufficient moisture for carbonation to occur. Therefore, it can be inferred that the carbonation phenomenon has no effect on the conservation of OC in drinking water (RH 100%), seawater (RH 100%), or the oven (RH 50%).

In drinking water and seawater, inorganic ions, such as Cl^−^, SO_4_^2−^, Mg^2+^, K^+^, and Na^+^, occur in aqueous media. According to Chen et al. [69], Mg^2+^ and SO_4_^2−^ may be more damaging to the compressive strength of cement-based composites than other anions. The major damage caused by SO_4_^2−^ to concrete is crystalline corrosion [6,70], which includes the conversion of thenardite crystals to mirabilite crystals [70,71] and the volume expansion of ettringite crystals and gypsum crystals. Both of the aforementioned factors may cause a sequence of physical and chemical reactions inside the concrete, which may result in additional concrete expansion and cracking [72,73,74]. Mg^2+^ damage is mostly indicated by the weakening of mortar and aggregate on the concrete’s surface [12,75]. This is due to Mg^2+^ generating magnetic Mg(OH)_2_ inside the concrete, which has low solubility. With the continuous precipitation of Mg(OH), C-S-H cementing material degrades and produces non-cementitious M-S-H [76,77]. These considerations support the RLs obtained during OC preservation in an aqueous medium. Furthermore, due to the high concentration of mineral salt in saltwater, the RL of OC preserved in seawater is larger than that in drinking water.

Previous research on the compressive behaviour of hardened concrete at various temperatures has been conducted. As the heating temperature increased, the compressive strength of concrete with the same moisture level decreased [78]. Concrete curing temperatures below 5 °C or over 100 °C resulted in an almost 20% drop in concrete strength [79]. When the gap between the ambient and concrete temperatures was small, the mechanical performance of concrete was ideal [80]. According to El-Zohairy et al. [81], after just 90 days, concrete lost 10–20% of its initial compressive strength when heated to 100 °C and 30–40% when heated to 260 °C. Therefore, in the current study, OC maintained in the oven had the lowest compressive strength (17.66 MPa). As a result, it displayed an RL value (52.30%) that was about twice as low as that of the OC after 28 days.

### 3.2. Wood–Cement Composite

#### 3.2.1. Untreated Redwood–Cement Concrete

Because of the nature of the untreated redwood chips, Figure 7 reveals an alarming rise in RL for all media. After 28 days, UW_I_ was characterized by excessive water absorption and increased swelling (7.8 MPa). In general, vegetated concrete has a high porosity. Indeed, the SEM pictures of a significant variety of fibre cements in the literature exhibit several fields when compared to regular concrete [82,83,84,85,86]. As a result, the addition of wood fibres to concrete enhances its density. In addition, because of their porosity, these concretes are water-sensitive [87]. 

After 12 months of storage in the open air, 36.92%, 19.23%, and 65.76% of RLs were obtained. Wood-fibre-concrete’s porous nature has a capacity to exchange moisture (or water) with the surrounding environment via the adsorption/desorption phenomena [14]. According to Mohr et al., the bulk of mechanical property losses occurred during the first five wet/dry cycles. However, ductile fibre failure was still identified using scanning electron microscopy (SEM) [88,89]. The sensitivity of vegetal concretes to humidity causes dimensional fluctuations that correspond to changes in relative humidity [90]. In addition, RH has been shown to influence at least four essential characteristics of electrospun fibres: surface morphology, interior porosity, structure, and mechanical qualities [91].

Wood-fibre–cement composites age in humid and aquatic environments, resulting in a reduction in strength and toughness. Wood composition in the specimens is often exposed to three distinct processes reducing their adherence to the matrix [41]:Debonding of the plant particle/matrix interfaces because of wood–water swelling.The degradation of plant particles is promoted by progressive alkali hydrolysis.Mineralization of plant particles is caused by the deposition of cement hydration products, primarily calcium hydroxide, onto the plant particle surface. On the other hand, the adhesion problem can be enhanced by modifying the particle surface.

The RL of UW_I_ in drinking water (19.23%) is insignificant compared to the non-aqueous environments after 28 and 365 days. In addition, The RL in drinking water was about 3.5 times more than the RL of UW_I_ stored in saltwater (65.76%). The pH of the storage media explains this finding. In fact, the pH of drinking water (pH 6) is lower than that of saltwater (pH 8–9), and the non-aqueous environments (pH 12–14). The lower the pH, the less alkaline the attack on the wood.

On the other hand, 68.84% of the RL was obtained after 12 months in the oven. Based on SEM images from the study by de Abreu Neto et al. [92], it is probable that increasing the temperature causes deformations and changes in the morphology of the wood fibres as a whole. Indeed, wood fibres exhibited a propensity for the thickness of their fibre walls to shrink with increasing temperatures. The study conducted by Dehghan et al. [93] supports this fact.

#### 3.2.2. Untreated Hardwood–Cement Concrete

Figure 7 depicts the variation of UW_II_ resistance according to the storage medium. Compared to the UW_I_ data, it is clear that the resistance improved considerably in all the selected environmental conditions. The discrepancy is explained by a difference in the absorption coefficient of the aggregates [12]. W_I_ has a lower apparent density than W_II_. However, this structure results in a material with poor strength and stiffness after cure. W_I_ has a very high water-absorption capacity because of its porous structure, which reduces the amount of water available for the binder to set in. Furthermore, the significant migration of water via capillarity results in the movement of Ca^2+^ calcium ions, which are required for binding the binder with the plant particles.

In order to verify this, the effect of the ratio of W_I_ and W_II_ on the density and adsorption capacity of UW_I_ and UW_II_ after 28 days of hardening was explored. Figure 8 demonstrates the outcomes. Figure 8a demonstrates that the densities of UW_I_ and UW_II_ reduce when the ratios of W_I_ and W_II_ in the concrete composition increase. This demonstrates that the wood chips from W_I_ and W_II_ lighten the concrete. Furthermore, Figure 8b shows that the absorption capacities of UW_I_ and UW_II_ are proportional to the rate of the wood chips. When the densities and water-absorption capacities of the two concretes are compared, it is evident that the densities and water-absorption capacities of UW_II_ are greater than those of UW_I_. This demonstrates that W_I_ has a lower density and a higher water-absorption capacity than W_II_.

According to Mukhopadhyay et al. [94], the degrees of deterioration caused by an alkaline environment attack on the vegetal additive’s composition vary. The loss of mechanical characteristics of various vegetables additives’ composition determines the extent of the damage. The severity of the attack ranges from a severe loss of strength to almost no loss of strength. It is governed by the capacity of the vegetal additive’s composition to absorb water.

The RG in all sectors, on the other hand, is lower than the OC. The result was that 15.67, 28.42, and 86.96% of RG were allocated when the UW_II_ samples were in the open air, potable water, and seawater. However, soaking UW_II_ in saltwater for 12 months yielded the best compressive strength results. Indeed, UW_II_ increased in its compressive strength by 40.5%. Even though multiple investigations have yielded equivalent results [72], the behaviour of concrete loaded with wood augmentation methods in saline solutions is still poorly understood. The rise in its compressive strength in seawater is due to the high concentration of inorganic salts. The concentration of inorganic ions in saltwater is greater than that in drinking water [95,96]. However, the competition for OH^−^ and Ca^2+^ attachment to the surface of wood chips with other inorganic ions is more crucial.

### 3.3. Treated Wood-Containing Concrete

#### 3.3.1. Cement Treatment of Wood

To improve some of the properties of the investigated compositions, such as water sensitivity and wood–matrix adhesion, treating the wood pieces before incorporating them into the concrete was considered. Various treatments have been used in previous works, and it has been demonstrated that coating with cement produces satisfactory results, particularly in terms of compressive strength. Because of this impact, W_I_ and W_II_ were treated with cement before being included in the concrete mixture. Figure 9 depicts the findings of the compressive strength of TW_IC_ and TW_IIc_, respectively, preserved in different media. The shaving treatment significantly enhanced the compressive strength of TW_IC_ and TW_IIc_ in all tested media, resulting in a suitably light structural weight. These findings are consistent with Bedirinan’s study [28].

Wood treatment lowers the porosity of TW_IC_ and TW_IIC_ structures, increasing the composites densities and decreasing the water capacities. This perspective is supported by Figure 10. It shows the findings of the change in the density and water absorption capacity of TW_Ic_ and TW_IIc_ as a function of the mass of W_I_ and W_II_ treated with cement and lime. When the equations of Figure 8a and Figure 10a are compared, the tangents of the density plots of TW_IC_ and TW_IIC_ are higher than those of UW_I_ and UW_II_. Similarly, the tangents of the water-absorption capacity graphs of TW_IC_ and TW_IIC_ are lower than those of U_WI_ and U_WII_ (Figure 8b and Figure 10b). These are the results of the wood-chip treatment. This finding demonstrates that treating wood chips with cement is an efficient way to create a more durable wood–cement composite.

In addition, it improves wood–matrix adhesion since the coating of the chips in a layer of cement is likely to reinforce the adhesion with the cementitious matrix. It is possible that the pre-coating of the chips minimizes the hygroscopic effect and increases the wood–matrix adhesion. The images of the SEM of treated and untreated wood chips (W_I_ and W_II_) in Figure 11 indicated that the cement matrix takes the shape of plates, and the distribution is random. They are more frequent in less inhabited places and specimens made of cement-treated chips. They can be seen clearly on the surface of the treated chips and do not exist on the untreated chips. These findings support the notion that the adhesion between wood and cement is poor. Furthermore, if the cement concentration in the treatment product is high, the paste layer coating the wood grains may be thicker, resulting in higher wood rigidity [72]. Consequently, significant compressive strength is attained in all of the investigated media.

#### 3.3.2. Mechanism of Cement–Wood Treatment

Based on Table 2, it is possible to conclude that the cement was constituted chiefly of lime (60.41%). Given this fact, W_I_ and W_II_ were subjected to lime treatment to understand better the influence of cement treatment chemistry on compressive strength. Figure 12 shows the results of TW_IL_ and TW_IIL_ compression strength in relation to the conservation medium. Figure 13 shows that TW_IL_ and TW_IIL_ were less resistive after 28 days than were TW_IC_ and TW_IIC_. The compression strengths of TW_IL_ and TW_IIL_ were 14.60 MPa and 15.05 MPa, respectively. The findings indicate that the strength of the concrete is only affected by the cement’s composition after 28 days in open air.

Figure 12 shows clear improvements in the specimens’ compression strength after 365 days of conservation in all environments investigated. This finding is supported by the fact that both TW_IL_ and TW_IL_ resistances were improved by 8.5 MPa compared to UW_I_ and UW_II_ resistances. The enhanced resistance was due to the lime’s ability to act as a surface layer for the wood chips [97]. This finding indicates that the improvement in compression strength of the TW_IC_ and TW_IIC_ was primarily attributable to the lime.

Figure 13 shows the findings regarding TW_IL_ and TW_IIL_ density and water-absorption capacity in relation to the mass of the wood chips. The tangents of the density plots of TW_IL_ compared to TW_Ic_, as well as TW_IIL_ compared to TW_IIC_, are almost identical. This means that the lime in the cement maintains the impermeability of water in the wood. However, TW_IL_ acquired resistance in all of the storage media except saltwater according to Table 7 (−24.59%). Similarly, besides the oven (−30.88%), all conservation zones in TW_IIL_ exhibited a grain in strength. The LRs collected are not related to the W_IL_ and W_IIL_ lime treatments.

The XRD patterns of W_II_, W_IIc,_ and W_IIL_ are shown in Figure 14. Table 8 and Table 9 present structural characteristics derived from XRD characterizations of W_I_ and W_II_, respectively.

The XRD data for W_I_ and W_II_ (Table 8 and Table 9) revealed 4 major peaks at 15.85°, 22.60°, 35.19°, and 45.49°. Those peaks could be attributable to C_10_H_8_O_4_ (27-1905), a derivative of cellulose crystal planes (C_6_H_10_O_4_) in wood fibres (W_I_ and W_II_) [98,99]. Thus, according to Camargo et al. [41], the major component of plant fibres is cellulose. In addition, no typical peaks at 16.85° and 22.60° were detected, as shown in Figure 14. A previous study of bamboo surfaces, investigated by Bao et al. [100], supports this fact. On the other hand, the XRD data of cement-treated wood (W_IC_ and W_IIC_), are differentiated by 3 peak ranges, namely: (i) 5 peaks at 47.32°, 48.62°, 51.74°,and 62.32° attribute to CaO_2_ (03-0865), which is derived from lime crystal (CaO) [5]; (ii) 7 peaks at 22.43°, 29.46°, 32.54°, 36.18°, 38.78°, 39.46°, and 49.90° attribute to SiO_2_ (82-1232); (iii) and 3 peaks at 34.34°, 41.35°, 43.53°, and 56.56° attribute to Al_2_O_3_ (42-1468).

In comparison to Table 2 and Table 3, the elements of cement with the largest weight percentages are CaO, SiO_2_, and Al_2_O_3_, in that order. They are also prominent components of clinkers, with FeO_2_, in low proportion, being the most abundant. This allows us to state that the cement has attached to the surface of the wood because the clinkers act as a glue. [101]. Similarly, there are 3 distinct sets of peaks in the lime-treated wood (W_IL_ and W_IIL_) spectra, namely: (i) 4 peaks at 28.84°, 29.58°, and 84.92° attribute to CaO_2_ (85-0514), (ii) 8 peaks at 18,19°, 34.25°, 47.33°, 50.94°, 54.49°, 62.76°, 64.49°, and 71.92° attribute to Ca(OH)_2_ (89-2779); and (iii) 4 peaks at 22.53°, 39.59°, 43.30°, and 48.68° attribute to CaCO_3_ (01-0837).

Based on these findings, it is possible to deduce that the adherence of the lime on the surface of the wood is only made by Ca(OH)_2_. The existence of CaCO_3_ on the surface of the wood is supported by the attack on Ca(OH)_2_ by humidity and CO_2_ in the air. According to Okayinkwa et al. [102], calcium carbonate significantly boosts the compressive strength of wood–cement concrete by decreasing its permeability. Furthermore, an SEM image of CaCO_3_/Wood–concrete, investigated by Merk et al. [103], shows that CaCO_3_ may be introduced deep into the wood structure using alternating solution–exchange cycles of extremely concentrated electrolytes NaHCO_3_ in water. Despite this fact, in the work conducted by Mejri et al. [104], it was demonstrated that the presence of sulphate and magnesium ions lowered the crystal development rate and decreased the quantity of CaCO_3_ precipitates obtained at a constant temperature and ionic strength. Magnesium caused the production of aragonite rather than calcite or vaterite. The presence of magnesium and sulphate ions caused changes in the effect of magnesium on the kinetics of CaCO_3_ precipitation, resulting in the incorporation of Mg^2+^ ions in the CaCO_3_ lattice. This explains the RL for TW_IL_ obtained after 365 days in sea water (Table 7).

On the other hand, according to Cheng Bencedira et al. [105], the increase in temperature increases CaCO_3_ precipitation in water (RH 100%). However, the CaCO_3_ precipitation depends on the percentage of RH in the medium. Indeed, according to Pondelak et al. [106], After one year of exposure at RH 33% and T = 45 °C, the CaCO_3_ does not precipitate. As a consequence, TW_IL_ loses 30.88% of its compression resistance. It is hypothesized that CaCO_3_’s stability is owing to the restricted quantity of water physiosorbed on the surface of the wood fibres, which is thought to be the driving factor for its precipitation.

According to Graine et al. [107], the average diameter of treated/untreated samples is calculated by measuring the width at half-height of the respective XRD peaks and by employing two methods, Scherrer’s and Williamson–Hall. Scherrer’s principle indicates that the widening of the XRD peaks is solely attributable to particle size. Using Equation (10), we can calculate the average crystallite size from the X-ray patterns:(10)$$D=\frac{K\lambda }{\beta_{corrected\;cos\theta }}\;$$
where *D* is the mean crystallite size, *K* is the crystallite form factor (12), is the X-ray wavelength, corrected is the full width at half the maximum of the peak, and is the diffraction angle. For correct calculations, the instrumental broadening effect and the observed XRD pattern peaks must be separated. As a result, Equation (11) can be used to estimate the corrected broadening, which is simply influenced by particle size.
(11)$$\beta_{corrected}\;cos\theta ={\left(\beta_{obseved}\;-\beta_{unstrumental}\right)}^{\frac{1}{2}}$$

However, the broadening of the XRD peaks might also be attributable to the lattice’s micro-strain impact. As a result, for greater precision, we must also separate the crystallite size effect and the micro-strain impact. In this paper, the Williamson–Hall approach was employed, which assumes Equation (12) where is ε  is the micro-strain in the lattice:


(12)
$$\beta_{corrected\;}cos\theta =4\varepsilon sin\theta +\frac{K\lambda }{D}$$


*D* variation is shown in Table 7 as 1.30 nm, 8.40 nm, and 7.55 nm by the Scherrer technique, and 1.83 nm, 21.55 nm, and 11.83 nm by the Scherrer method for W_I_, W_Ic_, and W_IL_, respectively, while, the variation of *D* for W_II_, W_IIc_, and W_IIL_ (Table 9) is 1.30 nm, 1.82 nm, and 8.34, respectively, according to the Williamson–Hall approach (Figure 15a) and 1.30 nm, 8.16nm, and 15.77 nm, respectively, according to the Scherrer method (Figure 15b). These data indicate that the wood chips’ Ds before treatment are the same. This supports the wood chip diameter results using granulometric analysis. In addition, the two approaches’ calculations demonstrate that the diameter of the wood particles increases when they are treated. In fact, *D* of cement-treated wood is larger than *D* of lime-treated particles. As a result, when the wood surfaces are treated, the lattice strain decreases. Indeed, the best lattice strain findings are attributable to W_IC_ and W_IIC_. The reduction in lattice strain is reflected in the reduction in spacing within the minimum wood width. As a result, the density of the wood rises, while its capacity to absorb water diminishes. The study conducted by Zhang et al. [108] supports this argument.

## 4. Conclusions and Limitations

This work contributes to the broader challenge of sustainable development by improving wood–concrete composites’ physical characteristics to expand their range of use. As a result, this work aimed to investigate the influence of various environments on the compression strength of concrete containing hardwood or redwood shavings. The studied storage environments were: open-air (T = 25 °C, RH 55%, pH 12–14), drinking water (T = 25 °C, RH 100%, pH 6–7), seawater (T = 25 °C, RH 100%, pH 7–9) and an oven (T = 75 °C, RH < 50%, pH 12–14). Due to the complex mechanisms involved in this study, only the compressive strength was experimentally measured. The main findings obtained are:-Ordinary concrete strength diminishes after 365 days of storage in every medium investigated compared to its strength after 28 days. Depending on the environment studied, this decrease is significant.-In all conditions investigated, the addition of wood (redwood or hardwood) to the concrete reduces its resistance. The low mechanical resistance of vegetable concretes is partly due to the high water-absorption of wood chips and the poor wood–matrix adhesion.-Concrete made of untreated hardwood is more resistant, in all preservation media, than concrete made of redwood due to its high apparent density.-Seawater enhances the strength of hardwood-based concrete. Indeed, compressive strength results for ordinary concrete and hardwood-based concrete are comparable.-The treatment of wood enhances the compressive strength after 365 days in all conservation media. The lime in cement acts on wood chips by (i) coating the surface of the wood, which lowers the porosity subtract of wood-based concrete, (ii) and reinforces the wood–matrix adhesion. The other cement constituents are only involved during the curing step (the first 28 days).

Even though treating wood with cement coating improves its resistance in watery environments, its resistance in dry and wet environments remains lower. The resistance of OC surpasses that of TW_IIc_ by 11.7 MPa, 12.51 MPa, and 8.86 MPa, respectively, after 28 days in the open air and 365 days in the open air and the oven. However, OC and TW_IIC_ resistance after 365 days in drinking water and saltwater are comparable. This enables us to conclude that the efficiency of cement treatment of wood for boosting compressive strength is restricted solely to watery conservation conditions. The research has implications for academics where the results of the study can be used to understand the wood–concrete composite material further. Some of the remedies to enhance its performance can include (a) describing the failure mode of each series using a variety of analytic methods, (b) changing the treatment of the surface of wood chips to linseed oil, (c) changing the type of surface treated, or (d) combining two forms of therapies.

## Figures and Tables

**Figure 1 materials-15-03572-f001:**
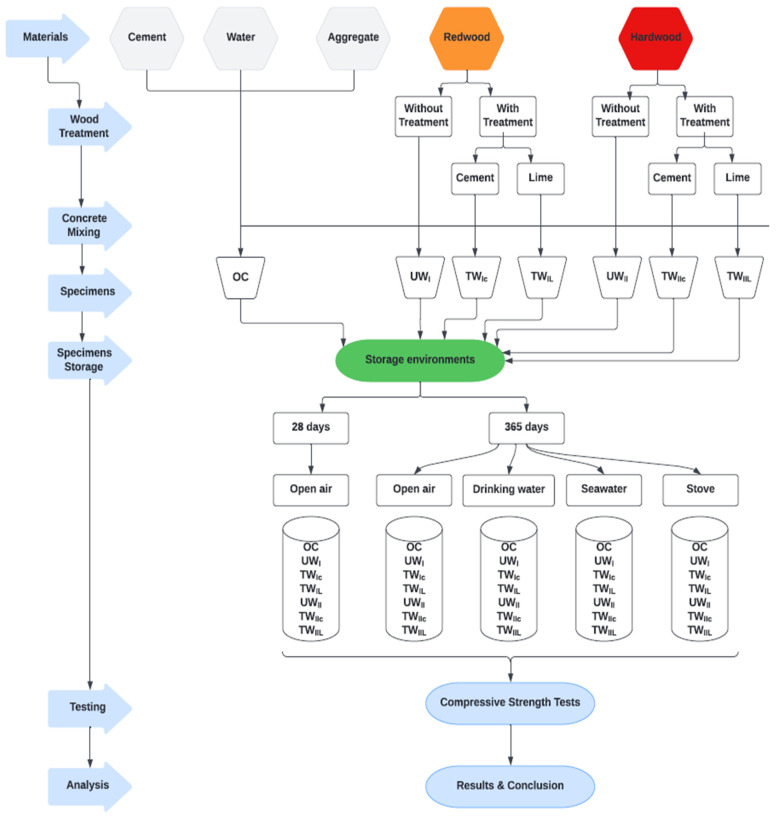
Methodological framework.

**Figure 2 materials-15-03572-f002:**
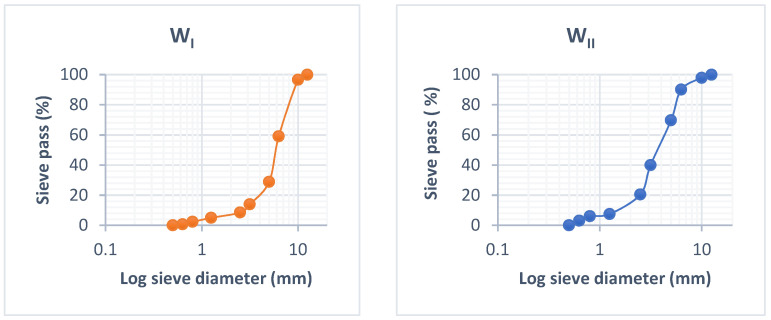
Wood (W_I_ and W_II_) chip granulometric curves.

**Figure 3 materials-15-03572-f003:**
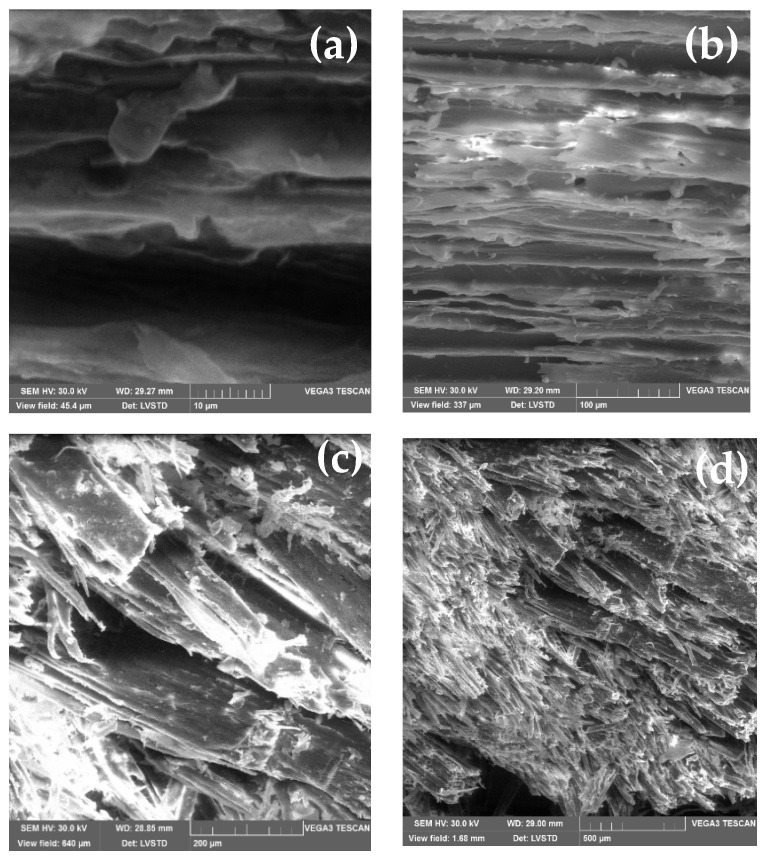
SEM microstructural images of W_I_ for various view fields: (**a**) 10 µm, (**b**) 100 µm, (**c**) 200 µm, and (**d**) 500 µm.

**Figure 4 materials-15-03572-f004:**
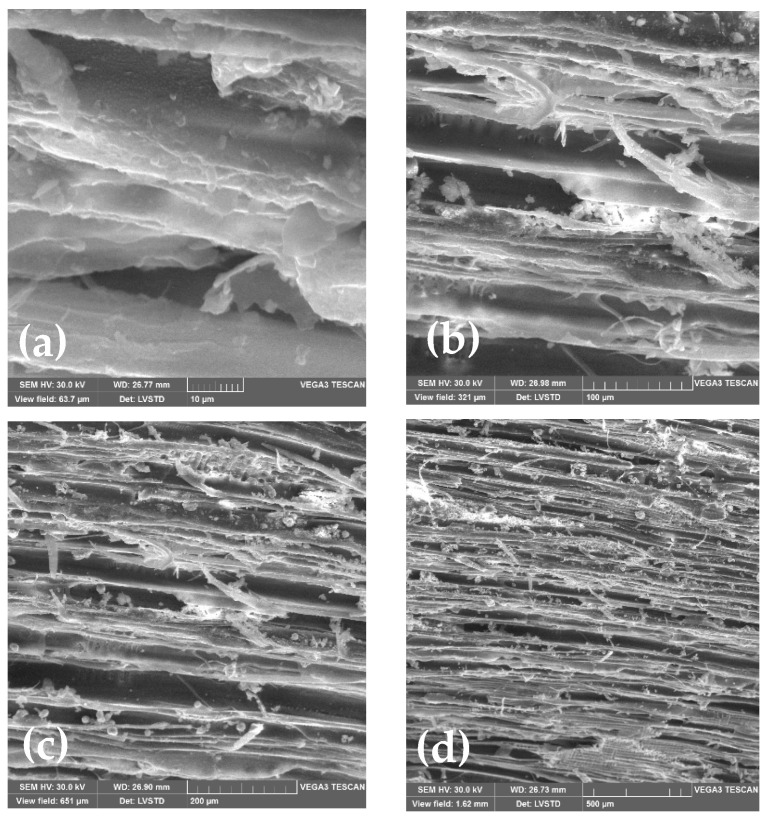
SEM microstructural images of W_II_ for various view fields: (**a**) 10 µm, (**b**) 100 µm, (**c**) 200 µm, and (**d**) 500 µm.

**Figure 5 materials-15-03572-f005:**
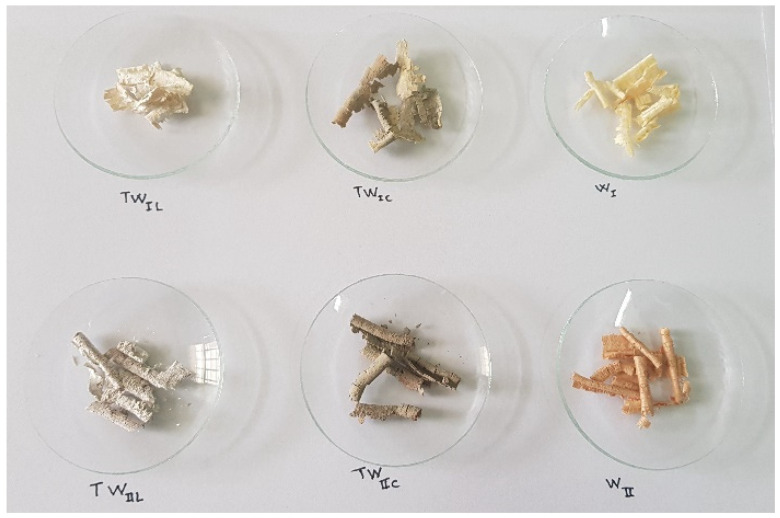
Image of W_I_ and W_II_ before and after treatment.

**Figure 6 materials-15-03572-f006:**
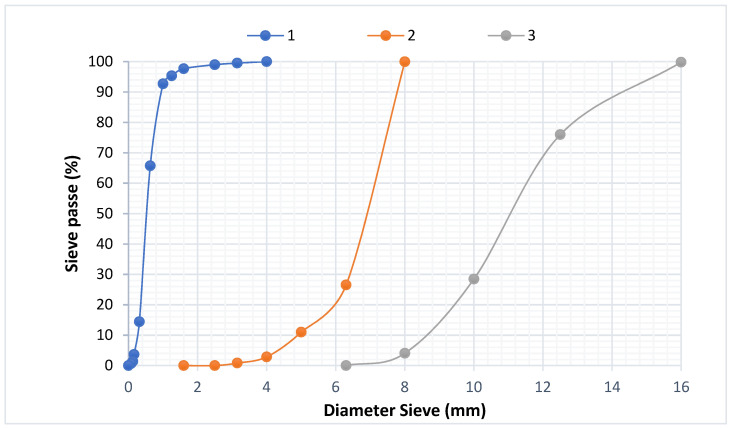
Granulometric curves and aggregate distribution graphs.

**Figure 7 materials-15-03572-f007:**
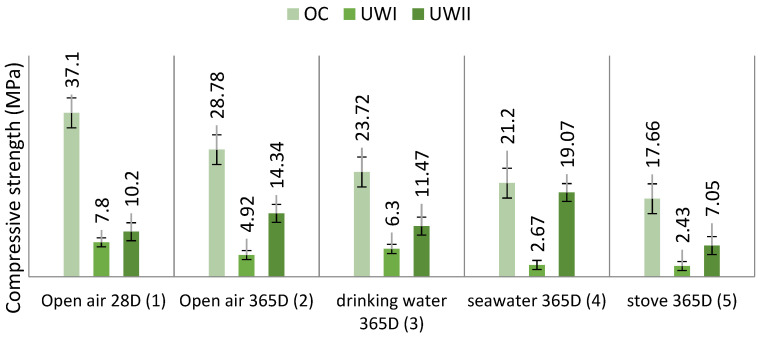
UW_I_ and UW_II_ compressive strength (in MPa) as a function of preserving media.

**Figure 8 materials-15-03572-f008:**
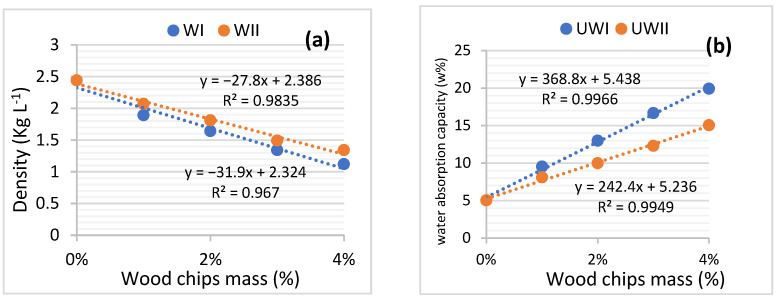
UW_I_ and UW_II_ (**a**) water-absorption capacity, and (**b**) density in function of wood chips’ mass.

**Figure 9 materials-15-03572-f009:**
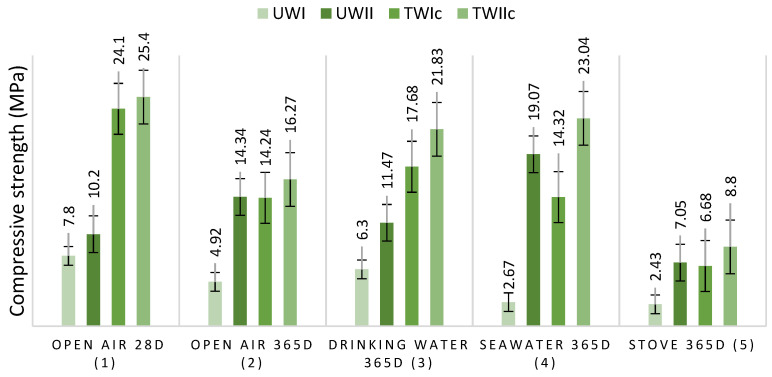
UW_I_, UW_II_, TW_IC,_ and TW_IIC_ compressive strengths as a function of preserving media.

**Figure 10 materials-15-03572-f010:**
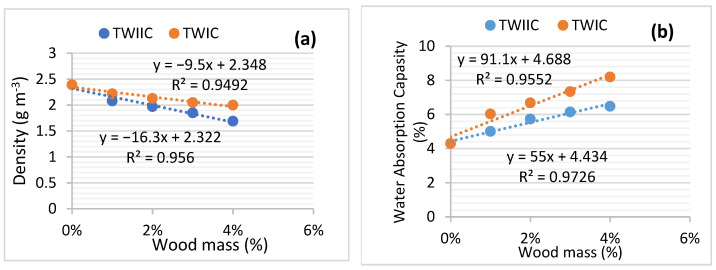
TW_Ic_ and TW_IIc_: (**a**) density, and (**b**) water-absorption capacity as a function of wood-chip mass.

**Figure 11 materials-15-03572-f011:**
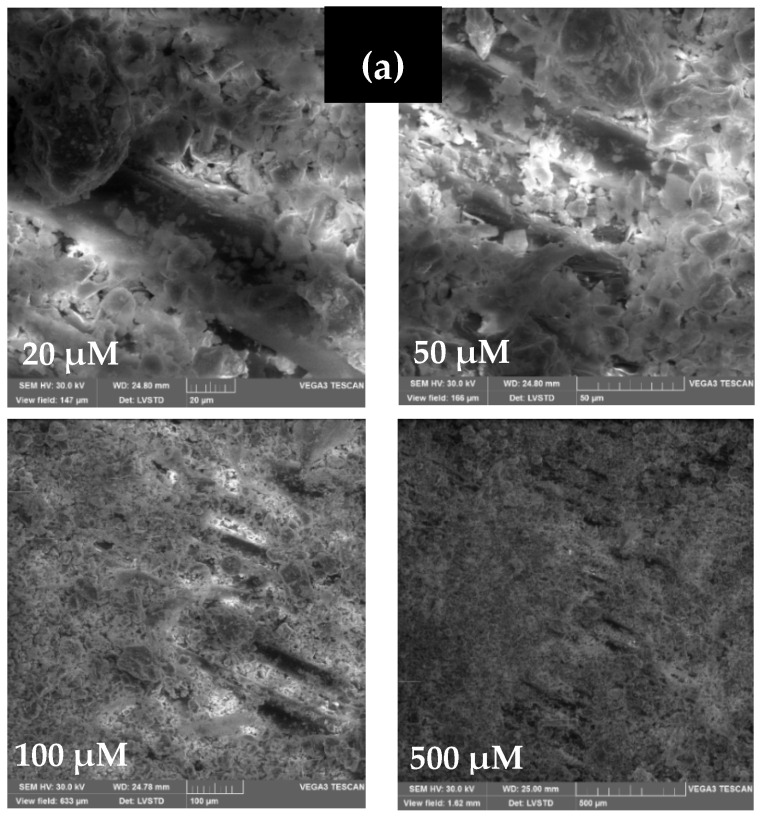
SEM microstructural images of (**a**) W_IC_ and (**b**) W_IIC_.

**Figure 12 materials-15-03572-f012:**
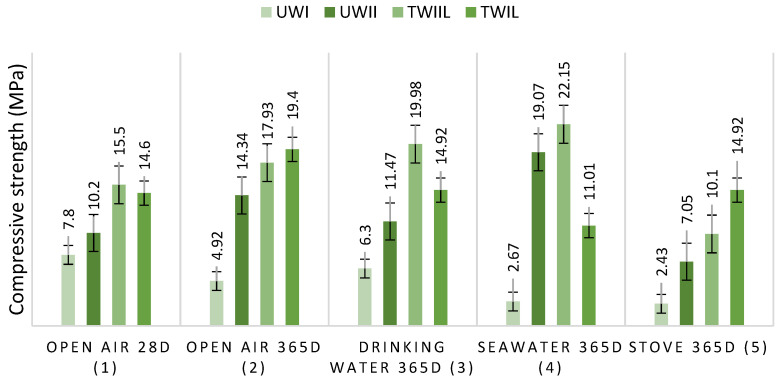
UW_I_, UW_II_, TW_IL,_ and TW_IIL_ compressive strength as a function of preserving media.

**Figure 13 materials-15-03572-f013:**
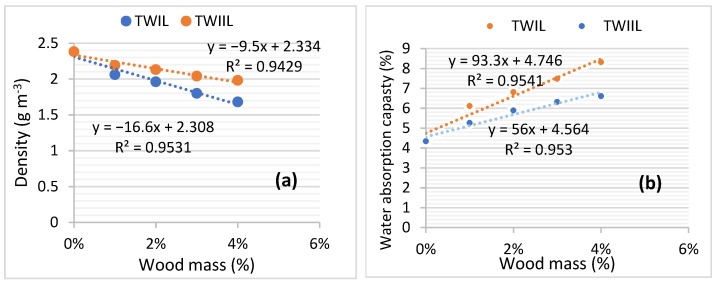
TW_IL_ and TW_IIL_ (**a**) density, and (**b**) water-absorption capacity as a function of wood-chip mass.

**Figure 14 materials-15-03572-f014:**
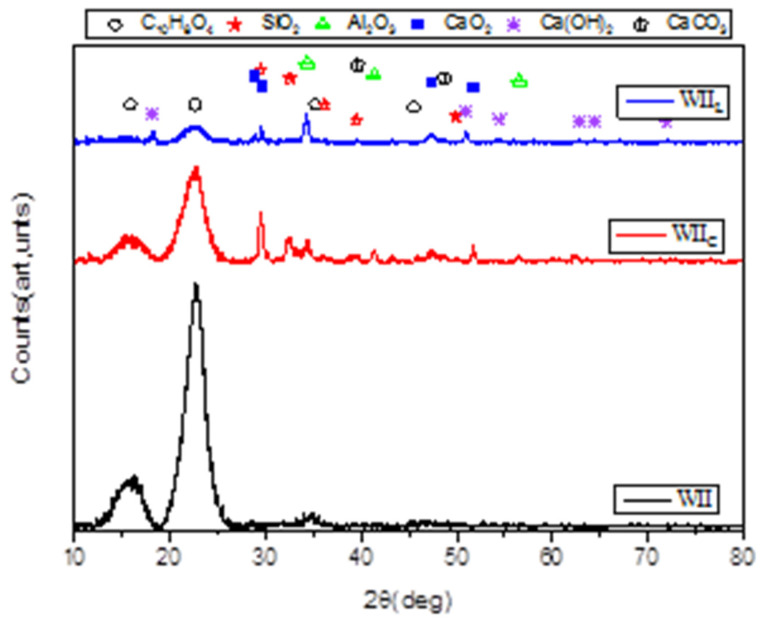
XRD spectra of W_II_, W_IIc_, and WI_IL_ samples.

**Figure 15 materials-15-03572-f015:**
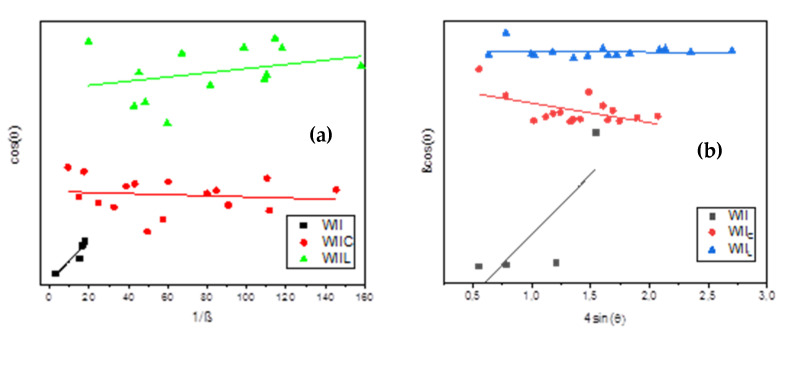
Particle size determination from (**a**) Scherrer’s and (**b**) Williamson–Hall plots derived from X-ray data for W_II._

**Table 1 materials-15-03572-t001:** Treatment methods for wood.

Treatment Methods	Advantage	Techniques	References
Thermal	Thermal modifications are the chemical and physical changes that occur in natural fibres because of temperature applications, where different process factors substantially impact fibre qualities.	Hornification Heat Hydrothermal	[44,45]
Chemical	These treatments eliminate contaminants from the surface of natural fibres, enhancing fibre–matrix adhesion.	Alkali Saline Formaldehyde	[25,46]
Coating	This reduces the impact of soluble compounds while increasing volume stability and surface roughness.	Linseed oil Cement Lime	[47]
Biological	Environmentally friendly fibre modification approaches, low-energy processing, softer reaction conditions, the ability to deploy recycling systems, and enhanced fibre characteristics were accomplished.	Enzymes Fungi Bacteria	[47,48]

**Table 2 materials-15-03572-t002:** Chemical composition of the Portland cement.

Elements	CaO	SiO_2_	Al_2_O_3_	Fe_2_O_3_	SO_3_	MgO
(%)	60.41	21.91	5.19	2.94	1.6	2.19

*(%): Weight percentage.*

**Table 3 materials-15-03572-t003:** Mineralogical composition of the Portland cement.

Elements	C_3_S	C_2_S	C_3_A	C_4_AF
(%)	58.2	18.5	9.3	8.2

*(%): Weight percentage.*

**Table 4 materials-15-03572-t004:** Mineralogical composition of the aggregates.

Properties	(1)	(2)	(3)	Standard
Apparent density (Kg m^−3^)	1380	1390	1360	NF P 18-554 et 18-555.
Absolute density (Kg m^−3^)	2600	2450	2450	NF EN 1097-3
Finesses Modulus	2.21	-	-	NF 18-540
Visual sand equivalent (%)	84.72	-	-	NF EN 933-8
d/D	0/5	3/8	8/16	-
Fragmentation resistance (%)	-	23	23	NF P 18-573
Wear resistance (%)	-	16	16	
Kurtosis (%)	-	8	8	NF P 18-561
Water Absorption (%)	-	0.2	0.2	NF P 18-554 et 18-555

**Table 5 materials-15-03572-t005:** Chemical composition of the aggregates.

Elements	CaO	SiO_2_	Al_2_O_3_	Fe	MgO	PF
(%)	54.70	0.11	0.45	0.12	null	43.74

**Table 6 materials-15-03572-t006:** Formulation of concrete.

Concrete	Cement (Kg m^−3^)	Water (Kg m^−3^)	E/C	(1) (Kg m^−3^)	(2) (Kg m^−3^)	(3) (Kg m^−3^)	Chips (Kg m^−3^)	Chips %
OC	400	208	0.55	626.53	168.68	927.25	-	0
UW	400	208	0.55	614	165.30	909.2	34.45	2
TW	400	208	0.55	614	165.30	909.2	34.45	2

**Table 7 materials-15-03572-t007:** RL/RG of TW_IIL_ and TW_IIL_ as a function of preserving media.

Environment	TW_IL_	TW_IIL_
28 days	0	0
Open air	32.88	40.59
Drinking Water	2.19	12.45
Seawater	−24.59	86.96
Oven	2.19	−30.88

**Table 8 materials-15-03572-t008:** Structural parameters obtained from XRD characterizations of W_I_, W_Ic,_ W_IL._

Wood Samples	Chemical Structure	JCPDS Card No	D (nm)		ε (µm)
			Scherrer Method	W–H Method	
W_I_	C_10_H_8_O_4_	27-1905	1.83	1.30	5.73 × 10^−2^
W_IC_	Al_2_O_3_	42-1468	21.55	8.40	5.60 × 10^−4^
	SiO_2_	82-1232			
	CaO_2_	03-0865			
W_IL_	CaCO_3_	01-0837	11.83	7.55	6.43 × 10^−4^
	Ca(OH)_2_	89-2779			
	CaO_2_	85-0514			

**
*D*
**
*:*
*Crystallite size (nm), **W–H**: Williamson-Hall, **ε**: Micro-strain.*

**Table 9 materials-15-03572-t009:** Structural parameters obtained from XDR characterizations of W_II_, W_IIc,_ W_IIL_.

Wood Samples	Chemical Structure	JCPDS Card No	D (nm)		ε (µm)
			Scherrer Method	W–H Method	
W_II_	C_10_H_8_O_4_	27-1905	1.83	1.30	5.73 × 10^−2^
W_IIC_	Al_2_O_3_	42-1468	8.168	1.82	9.36 × 10^−3^
	SiO_2_	82-1232			
	CaO_2_	03-0865			
W_IIL_	CaCO_3_	01-0837	15.77	8.34	5.15 × 10^−4^
	Ca(OH)_2_	89-2779			
	CaO_2_	85-0514			

**
*D*
**
*:*
*Crystallite size (nm), **W**–**H**: Williamson–Hall, **ε**: Micro-strain.*

## Data Availability

The data presented in this study are available upon request from the corresponding author.

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
