# Peer review of "Conservation Environments’ Effect on the Compressive Strength Behaviour of Wood–Concrete Composites"

_materials, 2022, doi:10.3390/ma15103572_

Round 1
Reviewer 1 Report
The authors have done an experimental investigation on the effect of treated/untreated wood chips on the strength development of concrete. Even the work is appreciable, the authors have to justify the following points to strengthen the paper:
- Include particle size distribution curve
- What about size of wood particles?
- Ratio of cement:water and lime:water for treatment purpose?
- How 35% reduction in the strength can be observed under outside air exposure? Is that mean that the concrete has lost 35% of its design strength within one year? Is it aggressive environment? Clarify this statement. An even in drinking water storage?
- If carbonation is the effect, has any study been carried out for carbonation?
- Similarly, in terms of untreated wood concrete, justifications were made for increased RL such as porosity, water absorption without any study been carried out on these.
- "WI has a lower apparent density than WII". Any proof?
- How SEM images of other study can be taken as justification for this work?
- Microstructural characterization such as SEM and XRD could be included to justify the results.
Author Response
We want to express our sincere gratitude to Reviewer #1 for the time dedicated to the review and the comprehensive, profound, and constructive remarks, which allowed us to improve the quality of our manuscript. The references are to the final line numbers of the revised article. In addition, the added or changed text of the manuscript was marked using “track changes” of Microsoft Word. We believe that this paper will be cited frequently by other authors. Kindly see the attachment (responses to reviewer 1).

Reviewer 2 Report
This paper is interesting and examines the effect on the compressive strength behavior of wood-concrete composites with treated/untreated wood chips.
Although adding wood chips to concrete is novel, it does not seem to work well. As the results obtained in this paper, the compressive strength decreases significantly. However, wood chip concrete may have many advantages in other ways, but the authors did not conduct other experiments.
Furthermore, many expressions in the article are unclear, which makes it difficult for readers to read.
Therefore, I believe that it cannot be published in the Materials in its present form.
Authors may consider the following comments for future work:
- The "Abstract" section is too long and needs to be shortened.
- Table 3 was tested by what method? there is a loss on ignition?
- Section 2.3.1, “they were immersed in the combinations (cement + water) and then (lime + water)” TWIL and TWIIL has been cement + water treated? Or just lime + water treated.
- Chemical formula 4 is wrong.
- What does the Y-axis of Figure 2?
- Why is the compressive strength of OC at 365 days lower than at 28 days? Usually, during water curing, the compressive strength will gradually increase due to the continuous hydration reaction of cement. Other maintenance methods also require detailed explanations.
- 295-296 “produce water (Eq. 7. Equation (8) depicts”
- Are chemical formulae (7) and (8) quoted correctly? It was not found in reference 27.
- The English of the article needs to be polished.
Author Response
We want to express our sincere gratitude to Reviewer #2 for the time dedicated to the review and the comprehensive, profound, and constructive remarks, which allowed us to improve the quality of our manuscript. The references are to the final line numbers of the revised article. In addition, the added or changed text of the manuscript was marked using “track changes” of Microsoft Word. We believe that this paper will be cited frequently by other authors. Kindly see the attachment (responses to reviewer 1).

Reviewer 3 Report
The authors investigated the long-term stability of different wood- concrete composites under different conditions. This work is important in term of durability estimation, and the effects of the wood treatment on the compressive strength of the composites. The manuscript fits well with the scope of the journal, and it is important research. The weak points of this work are using only one factor to measure the mechanical behavior of the specimens, i.e. compressive strength, poor description of the specimens and testing ….
General comments:
- Add more details about the wood: composition, sizes …etc.
- Can you describe the failure mode of each series?
- Can you provide images of the samples before or after the testing?
- Can you show the error bars of each test, how many samples were tested for each series?
- Show the error bars of each test, how many samples were tested for each series?
- The title
- I suggest shortening the title to:” Conservation environments’ effect on the compressive strength behavior of wood-concrete composites”
- Line 22: “a resistance loss of 35.84, 36.06, 42.85, and 52.30 % were observed respectively.”
- Resistance loss= Compressive strength decreased?
- 36%, 36%, 43%, and 52%
- Section 2.3
- What are the dimensions of the specimens?
- Section 2.3.1
- Further clarification is required on wood treatment process
- Line 214: “The compressive strength loss (RL) and gain (RG) are computed using the formulae (1) 214 and (2), respectively.”
- May be you can use only one equation (i.e. strength loss), so any strength loss less than 0 is considered as strength gain.
- Figure 2
- This figure is not necessary because the results of OC compressive strength are already shown in Figure 3
- Line 212: “Instron hydraulic press built in the Soviet Union, model 212 ZIM (M) type n-10, N° 4577”
- Are you sure that the Instron hydraulic press was made in the Soviet Union?
- Tables 2 and 3 …
- Did you measure these results? If not, please mention the reference/s in the caption.
Author Response
We want to express our sincere gratitude to Reviewer #3 for the time dedicated to the review and the comprehensive, profound, and constructive remarks, which allowed us to improve the quality of our manuscript. The references are to the final line numbers of the revised article. In addition, the added or changed text of the manuscript was marked using “track changes” of Microsoft Word. We believe that this paper will be cited frequently by other authors. Kindly see the attachment (responses to reviewer 1).

Round 2
Reviewer 1 Report
Thanks for incorporating the comments
Author Response
We want to express our sincere gratitude to Reviewer #1 for the time dedicated to the review and the comprehensive, profound, and constructive remarks, which allowed us to improve the quality of our manuscript.
Reviewer 2 Report
This paper is revised.
Author Response
We want to express our sincere gratitude to Reviewer #2 for the time dedicated to the review and the comprehensive, profound, and constructive remarks, which allowed us to improve the quality of our manuscript.
Reviewer 3 Report
The authors have satisfactorily addressed most of my concerns. The only remaining concern is that there are many illustrations. So I suggest I combining figures with similar process such as figures12 & 13, 17 & 18 ... etc. Also I suggest deleting less important Figures such as 7, 8, 9, 10 .... the same can be done with the tables.
Regarding the references of table 2 and 3: if this data was not published elsewhere then you can delete these references which refer to technical lab.
The only remaining concerns I have with the abstract are seemingly minor:
Author Response
We want to express our sincere gratitude to Reviewer #3 for the time dedicated to the review and the comprehensive, profound, and constructive remarks, which allowed us to improve the quality of our manuscript. The added or changed text of the manuscript was marked using “track changes” of Microsoft Word. We believe that this paper will be cited frequently by other authors. Kindly see the attachment (responses to reviewer 3).
